# Clear Cell Carcinoma of the Abdominal Wall as a Rare Complication of General Obstetric and Gynecologic Surgeries: 15 Years of Experience at a Large Academic Institution

**DOI:** 10.3390/ijerph16040552

**Published:** 2019-02-14

**Authors:** Yen-Ling Lai, Heng-Cheng Hsu, Kuan-Ting Kuo, Yu-Li Chen, Chi-An Chen, Wen-Fang Cheng

**Affiliations:** 1Department of Obstetrics and Gynecology; National Taiwan University Hospital, Taipei 100, Taiwan; adina@kimo.com (Y.-L.L.); chianchen@ntu.edu.tw (C.-A.C.); wenfangcheng@yahoo.com (W.-F.C.); 2Graduate Institute of Clinical Medicine, College of Medicine, National Taiwan University, Taipei 100, Taiwan; b101092037@gmail.com; 3Department of Obstetrics and Gynecology, National Taiwan University Hospital Hsin-Chu Branch, Hsin-Chu 300, Taiwan; 4Department of Pathology and Graduate Institute of Pathology, College of Medicine, National Taiwan University, Taipei 100, Taiwan; kuokt@ntu.edu.tw; 5Department of Graduate Institute of Oncology, College of Medicine, National Taiwan University, Taipei 100, Taiwan

**Keywords:** endometriosis, malignant transformation, clear cell carcinoma, abdominal wall, inguinal lymph node, cesarean section

## Abstract

The objective of this article was to report the clinicopathological characteristics, treatment modalities, and outcomes of patients with clear cell carcinoma (CCC) of the abdominal wall. Medical records of six patients diagnosed with CCC of the abdominal wall between May 2003 and May 2018 at the National Taiwan University Hospital were reviewed. All patients had prior obstetric or gynecologic surgeries. The primary clinical presentation was enlarging abdominal masses at previous surgical scars. Four patients underwent initial/primary surgeries with/without adjuvant chemotherapy. One patient received neoadjuvant chemotherapy followed by surgical intervention and adjuvant chemotherapy, the other received chemotherapy and sequential radiotherapy without any surgical intervention. Two of four patients undergoing initial/primary surgeries had disease recurrence and the remaining two cases without initial surgery experienced disease progression during primary treatment. Inguinal lymph nodes were the most frequent sites of recurrence. In conclusion, previous obstetric or gynecologic surgery can be a risk factor for CCC of the abdominal wall. Complete resection of abdominal wall tumor and suspected intra-abdominal lesions with hysterectomy and bilateral inguinal lymph nodes dissection may be the primary treatment. Adjuvant chemotherapy would be considered for potential benefits. For patients without bilateral inguinal lymph nodes dissection, careful inguinal lymph node palpation during postoperative surveillance is necessary. More cases are still needed to elucidate the clinical management of this disease.

## 1. Introduction

It is estimated that cancers arise in approximately 1% of woman with ovarian endometriosis [1]. Clear cell and endometrioid carcinomas are the malignancies most commonly seen in ovaries containing endometriosis [2]. Malignant transformation of extra-ovarian endometriosis is rare and the most frequently reported sites are the rectovaginal septum, colon, and vagina [1]. Implantations of ectopic endometrium in the scar tissue of the anterior abdominal wall after operations, especially cesarean sections, laparoscopic surgeries, and hernia repairs, is referred to as abdominal wall endometriosis [3]. It is plausible that abdominal wall endometriosis could undergo malignant changes resulting in clear cell carcinoma (CCC) of the abdominal wall [4]. CCC originating in the abdominal wall is an extremely rare form of cancer. After the first case documented in 1986 [5], only few cases have been reported to date.

In 1925, Sampson described the first case of malignant transformation of ovarian endometriosis, and defined three criteria for the diagnosis of malignant transformation of endometriosis: (1) Demonstration of both neoplastic tissue and endometriosis within the tumor, (2) histological appearance resembling endometrial stroma surrounding characteristic glands, and (3) no other primary tumor site [6]. In addition, Scott added the fourth criterion of metaplasia between benign endometriosis and carcinoma in 1953 [7]. However, about 44% of reported cases did not correlate well with all these four histological criteria [8,9,10,11,12,13,14,15].

The management of CCC of the abdominal wall is not well established because of its extremely low incidence rate. Therefore, we comprehensively reviewed the characteristics of six cases with this disease in our institution over a 15-year time period. To our knowledge, the current case series is the largest one to date. It is noteworthy that this is the first study reporting the recurrence pattern of CCC of the abdominal wall, which may have potential to modulate the primary treatment strategy for the improvement of clinical outcomes for these patients.

## 2. Materials and Methods

Following approval by the National Taiwan University Hospital Institutional Review Board, a retrospective review was conducted (201805083RIND). All medical records of 6 patients treated for CCC of the abdominal wall in the National Taiwan University Hospital between May 2003 and May 2018 were reviewed. These cases were confirmed via pathological diagnosis. The medical parameters, including basic characteristics, primary treatment modalities, pathologic characteristics, and clinical outcomes, were comprehensively reviewed.

The basic characteristics included age at diagnosis, body mass index (BMI), parity, previous obstetric and gynecologic surgery histories, the interval between the most recent obstetric or gynecologic surgeries and disease diagnosis, tumor size, presenting symptoms, pretreatment serum cancer antigen 125 (CA-125) level, and findings of pretreatment computed tomography (CT). The treatment strategies included the methods of surgery, timing of administration of chemotherapy, and use of radiotherapy. The pathologic characteristics included histology, status of surgical margin, co-existing endometriosis, and existence of granuloma caused by suture materials of the abdominal wall tumor. When the removal of gynecologic organs was performed, pathologic examination for CCC and endometriosis in these specimens were recorded.

The clinical outcomes included recurrent sites, types of salvage treatment, disease-free survival (DFS), overall survival (OS), and current status of these patients. Periodic examinations during follow-up included history-taking, pelvic and rectal examinations, and regional lymph node palpation every 3 months for 3 years, and every 6 months thereafter after completion of the primary treatment. Levels of serum CA-125 was determined on each visit. Smears of the vaginal cuff were done annually. CT or magnetic resonance imaging (MRI) was done for suspected recurrence. CA-125 levels ≥2-fold the upper limit of normal in two consecutive tests with 2-week intervals, abnormal results of imaging studies or tissue proven from a biopsy was considered as recurrence. DFS was defined as the length of time after completion of the primary treatment until the date of disease relapse or last follow-up. OS was calculated as the time from initial diagnosis of CCC of the abdominal wall until the date of death or last follow-up.

## 3. Results

### 3.1. Prior Obstetric or Gynecologic Surgery with the Abdominal Wall Contacted with the Endometrium or Endometriotic Tissue may be a Risk Factor for the Development of CCC of the Abdominal Wall

Table 1 shows the clinicopathologic characteristics of the six patients. The mean patient age at the time of diagnosis was 52.7 years. The mean BMI was 22.6 kg/m^2^ (range, 17.0–28.8 kg/m^2^). All patients received obstetric or gynecological surgeries. Five patients had cesarean section histories and one had laparoscopic surgery for ovarian endometriosis. The mean time from the most recent obstetric or gynecologic surgery to diagnosis of CCC of the abdominal wall was 20.2 years (range, 4.0–33.0 years). The shortest interval (4.0 years) was noted in the patient with laparoscopic surgery (case 3). The mean tumor size was 10.1 cm (range, 4.8–17.5 cm). Five patients with cesarean section histories presented with an enlarging lower abdominal wall mass containing scars of a cesarean section (Figure 1A), and the other patient complained of an enlarging left lower quadrant mass near a laparoscopic trocar wound. The mean pretreatment serum carcinoma antigen 125 (CA-125) level was 28.3 U/mL (range, 20.1–38.9 U/mL; normal value < 35.0 U/mL).

### 3.2. Modalities of Primary Treatment

The primary treatment strategies of six patients for CCC of the abdominal wall are demonstrated in Table 1. One patient received tumor excision plus total abdominal hysterectomy (TAH) and bilateral salpingo-oophorectomy (BSO) without adjuvant treatment (case 1). Three patients received tumor excision plus intra-abdominal surgeries (case 2, TAH, BSO, bilateral pelvic lymph node dissection (BPLND), and infracolic omentectomy; case 3, TAH and BSO; case 4, TAH, BSO, and BPLND) followed by adjuvant chemotherapy. The regimen of adjuvant chemotherapy was paclitaxel (175 mg/m^2^) and carboplatin (AUC = 5 or 6).

Case 5 initially underwent two regimens of neoadjuvant chemotherapy: (1) Paclitaxel (175 mg/m^2^) and carboplatin (AUC = 5) combined with bevacizumab (7.5 mg/kg), and (2) gemcitabine (800 mg/m^2^) and carboplatin (AUC = 5). Surgical intervention with tumor excision, TAH, BSO, and infracolic omentectomy was performed for tumor progression. Postoperative adjuvant chemotherapy was prescribed using gemcitabine (800 mg/m^2^) and carboplatin (AUC = 5) combined with bevacizumab (7.5 mg/kg). However, the disease progression was still noted. Case 6 received tumor biopsy other than resection for disease diagnosis. This patient received two regimens of chemotherapy: (1) paclitaxel (175 mg/m^2^) and carboplatin (AUC = 5), and (2) liposomal doxorubicin (40 mg/m^2^) and carboplatin (AUC = 5). In addition, sequential radiotherapy with a total dose of 4400 cGy was administered after completing primary chemotherapy. However, the tumor progression still could not be controlled.

### 3.3. Malignancies without Clinical Manifestations of Image Studies are not Pathologically Confirmed in Pelvic Organs

The pathologic characteristics of surgical specimens are shown in Table 1. The histology of the abdominal wall tumors was confirmed as CCC (Figure 1B). Co-existing endometriosis in these abdominal wall tumors was identified in two cases (case 3 and case 5). In addition, granulomas caused by suture materials could be found in two tumor tissues (case 1 and case 4) in the histological examination (Figure 1C). Free margins with a distance more than 2 cm from the abdominal wall tumors were noted in four cases (case 1–4), but the surgical margin was involved with tumor cells in case 5.

In addition to the resection of the abdominal wall tumors, the removal of gynecologic organs, including uterus, bilateral ovaries, and fallopian tubes, was performed in five patients (case 1–5, Table 1). Intra-abdominal endometriosis by pathologic examination was found in three cases (case 1, 3, and 5), but results for malignancy were all negative in the gynecologic organs. The pathologic findings were compatible with those of preoperative CT scans, which showed abdominal wall tumors without the involvement of gynecologic organs (Table 1, Figure 1D,E). However, in case 2, the preoperative CT scans demonstrated right pelvic lymphadenopathy (Figure 1F), which was pathologically confirmed as metastasis of the right pelvic lymph nodes.

### 3.4. Inguinal Lymph Nodes are the Most Common Site of Disease Relapse

Clinical outcomes of the six patients are shown in Table 2. After completing primary treatment, three patients had a period of DFS (case 1 = 10.0 months; case 2 = 3.0 months; case 3 = 93.0 months). No evidence of disease relapse was detected in case 3. One patient (case 4) was under the treatment of postoperative adjuvant chemotherapy. The remaining two patients (case 5 and 6) experienced progressive enlargement of abdominal wall tumors even under the different primary treatment modalities.

Sites of tumor recurrence and metastasis are shown in Table 2 and Figure 2. The inguinal lymph node (Figure 2A) was the most common site of disease recurrence and progression (case 1, 2, 5, and 6). Other metastases, including bone (Figure 2B, case 1), neck lymph node (Figure 2C, case 5), liver (Figure 2D, case 5), and lung (Figure 2E, case 5), were also observed. In addition, the management of disease relapse is recorded in Table 2. In case 1, disease recurrence was detected in the inguinal lymph node and bone. She did not receive any salvage treatment because the disease progressed rapidly and her hepatic function was poor and caused by severe fatty liver disease. In case 2, tumor relapse was only located in the inguinal lymph node. Resection of this lesion with a free margin was performed. No evidence of disease was noted during follow-up.

In case 5, the progression of the abdominal wall tumor with metastasis to the inguinal lymph nodes was noted during the neoadjuvant chemotherapy. Then, resection of this progressive lesion with the involved margin was performed. Multiple distant metastases, including neck lymphadenopathy, liver, and lung, were detected while receiving adjuvant chemotherapy. This patient received salvage chemotherapy and radiotherapy, but died with OS 23 months. In case 6, the abdominal wall tumor progression with inguinal lymphadenopathy was noted under chemotherapy and sequential radiotherapy. However, no salvage treatment was arranged for progressive disease because of the poor condition of this patient. The mean OS of these six patients was 26.1 months, with a range of 5.0 to 97.0 months.

## 4. Discussion

CCC of the abdominal wall is extremely rare. Taburiaux et al. reviewed existing reports in the English language literature on cancer arising from abdominal wall endometriosis from September 1986 to August 2014 [4]. A total of 27 cases were identified, and 18 patients were reported to have clear cell carcinomas. Our retrospective review described six cases of pathologically confirmed clear cell carcinoma of the abdominal wall in detail over 15 years at our institution. In this study, previous obstetric or gynecologic surgery having abdominal wall contact with the endometrium or endometriotic tissue may be a risk factor for the development of CCC of the abdominal wall. Tumor resection with a free margin may be the essential treatment modality for this disease. The inguinal lymph nodes were the most common location of disease recurrence.

In our case series, all patients had a history of cesarean section or laparoscopic surgery for endometriosis, consistent with previous reports [5,8,9,10,11,12,13,14,15,16,17,18,19,20,21,22,23]. A previous surgical procedure with endometrial cavity opening or endometriosis may be a risk factor for developing CCC of the abdominal wall, which could be supported by the granulomas caused by suture materials found in tumor samples (Figure 1C). Dissemination of endometrial tissue or endometriosis at the time of surgery is biologically plausible because there is an opportunity for the inoculation of endometrial cells from hysterotomy or endometrioma to the peritoneum or abdominal wall [3,4].

Several studies have demonstrated that endometriosis can be the precursor and have the potential for malignant transformation to CCC of the ovaries [2,24,25,26]. In the tumors coexisting with endometriosis, a benign endometriotic gland can be observed to merge with atypical and overtly malignant glands [6,24]. Taburiaux et al. reported that 56% of patients with CCC of the abdominal wall had co-existing endometriosis. In our series, only two patients had co-existing endometriosis found in the abdominal wall tumors (case 3 and case 5). Underreporting of the association of malignancy with endometriosis could be attributed to several factors [1]. First, the sampling technique may not be adequate to find a small focus of endometriosis adjacent to a malignant tumor. Second, a cancer may destroy the endometriotic tissue from which it arose. Third, the endometriosis may be considered to be a minor component. Not all CCC of the abdominal wall were endometriosis-associated malignancies. Therefore, the elucidation of the pathogenesis is essential for the prevention and treatment of this disease.

The mean interval between the last surgery and the diagnosis of CCC of the abdominal wall was long (20.2 years), which suggested a slow evolution of the tumorigenesis (Table 1). The typical manifestation was an abdominal wall mass adjacent to previous surgical scars. All the serum CA-125 levels were not greater than 40 U/mL in our case series, suggesting that CA-125 may not be an appropriate tumor marker for CCC of the abdominal wall.

Primary wide tumor resection combined with total abdominal hysterectomy and bilateral salpingo-oophorectomy or cancer staging surgery was the preferred modality of primary treatment at our institution. For complete removal of all visible tumor tissue, a synthetic mesh or flap may be required for abdominal wall reconstruction. All the specimens without malignant suspicion on preoperative CT scanning histologically showed no evidence of clear cell carcinoma. Only right pelvic lymphadenopathy on preoperative images was finally confirmed as metastasis by pathologic examination (Figure 1F). Thus, when considering surgical management, grossly total tumor resection may be appropriate for diagnostic and therapeutic purposes. As for gynecologic organs, given the fact that endometrium contact may be the risk factor for CCC of the abdominal wall, it still can be reasonable to perform hysterectomy routinely even though no evidence of malignancy in gynecologic organs was detected in our cohort.

Disease control of primary surgery with or without adjuvant chemotherapy was good. In our study, 50% of patients (3/6, case 1–3) had disease remission after primary operation. One patient (1/6, case 4) without evidence of disease was under postoperative chemotherapy. Two patients (2/6, case 5 and 6) who did not receive initial tumor resection with a free margin experienced uncontrolled disease progression. Therefore, initial complete tumor resection with/without adjuvant chemotherapy (case 1–4) seemed to have better disease control in comparison with other primary treatment modalities, such as neoadjuvant chemotherapy followed by operation and adjuvant chemotherapy (case 5), or chemotherapy and sequential radiotherapy (case 6). The role of adjuvant chemotherapy was not established. In our cohort, one patient not receiving adjuvant treatment eventually encountered disease recurrence (case 1). Four patients received adjuvant chemotherapy (case 2–5), two of them remained without evidence of disease (case 3 and 4), and the other two experienced disease recurrence (case 2 and 5). Overall, it would be difficult to draw a conclusion based on such limited cases. However, the results seemed to support the potential benefits of adjuvant chemotherapy. Therefore, administration of adjuvant chemotherapy after primary operation could be considered. At our institution, platinum-based chemotherapy was the most used.

The survival intervals and prognostic factors are difficult to investigate due to the limited number of cases with standard management to treat CCC of the abdominal wall [27]. Taburiaux et al. reported a median survival time of 30.0 months [4]. In our study, the mean OS of the studied population was 26.1 months. In addition, no associated risk factor for poor prognosis can be identified because of the disease rarity.

Patterns of disease recurrence/metastasis have not been reported in the literature before. The most common site of recurrence and metastasis was the inguinal lymph node. Inguinal lymphadenopathy can be detected in all cases with disease recurrence or progression (Table 2). The reason may be that superficial lymphatics of the abdominal wall located below the umbilicus run in an inferior direction towards the superficial inguinal lymph nodes [28]. Other recurrent and metastatic locations, including bone, neck lymph node, liver, and lung, were also noted in certain cases. Considering the effective treatment modality and common recurrent/metastatic site of CCC of the abdominal wall, complete resection of the abdominal wall tumor and intra-abdominal suspected lesions on preoperative image studies with hysterectomy and bilateral inguinal lymph node dissection may be suggested as the first step to treat these patients (Figure 3).

## 5. Conclusions

CCC of the abdominal wall is a rare and distinct entity. Our study addressed comprehensive clinical courses and outcomes of these cases in a single institution. Complete resection of the abdominal wall tumor and intra-abdominal suspected lesions on preoperative image studies with hysterectomy and bilateral inguinal lymph node dissection may be suggested as the first step to treat CCC of the abdominal wall. Adjuvant chemotherapy would be recommended for potential benefits. For those patients without bilateral inguinal lymph node dissection, careful inguinal lymph node palpation during postoperative surveillance is needed. More such cases are still needed to elucidate the survival benefits of this surgical procedure, the role of adjuvant chemotherapy or radiotherapy, and the protocol of disease follow-up.

## Figures and Tables

**Figure 1 ijerph-16-00552-f001:**
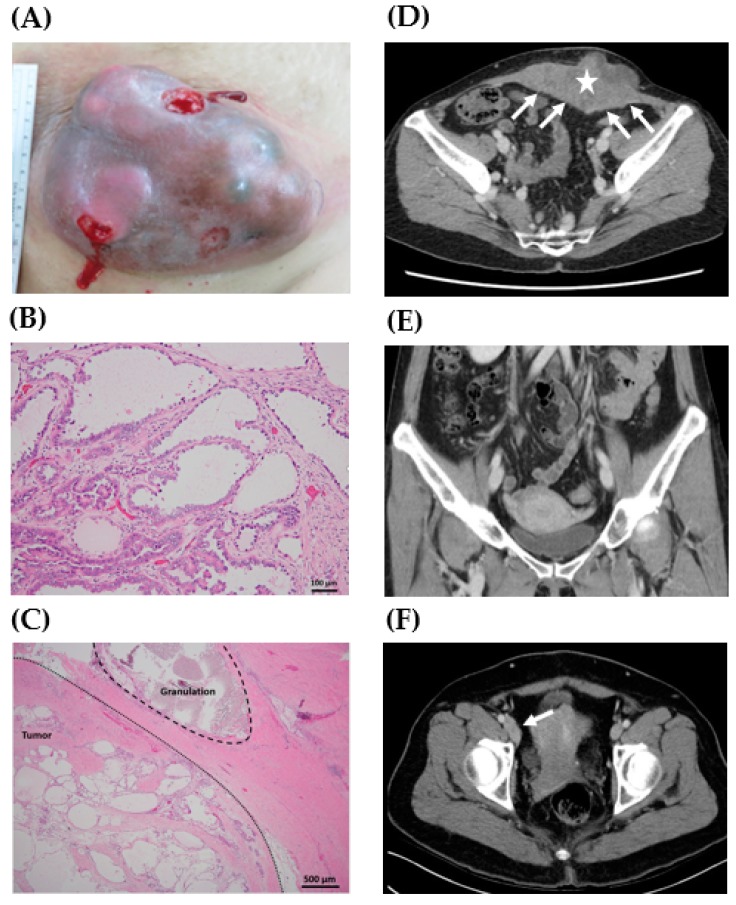
Representative gross, histology, and preoperative computed tomography (CT) scan of clear cell carcinoma of the abdominal wall. (**A**) A 17.5 cm mass lesion in the lower abdomen with an ulcerative surface (case 1). (**B**) Pathologic examination of the abdominal wall tumor showed tubulocystic growth patterns lined by cuboidal, hobnail cells, and clear cells. Focal papillary, micropapillary, and cribriform patterns were also present (case 4, H&E stain, 200X). (**C**) Granuloma caused by suture material was noted in the tumor sample (case 4, H&E stain, 40X). (**D**) A 13.0 cm lobulated heterogeneous tumor (star) was located at the anterior lower abdominal wall with peritoneal involvement (arrows) (case 4). (**E**) No definite lesions in the uterus, ovaries, and fallopian tubes were noted (case 4). (**F**) Lymphadenopathy along the right inferior epigastric vessels and external iliac vessels was noted (arrow) (case 2).

**Figure 2 ijerph-16-00552-f002:**
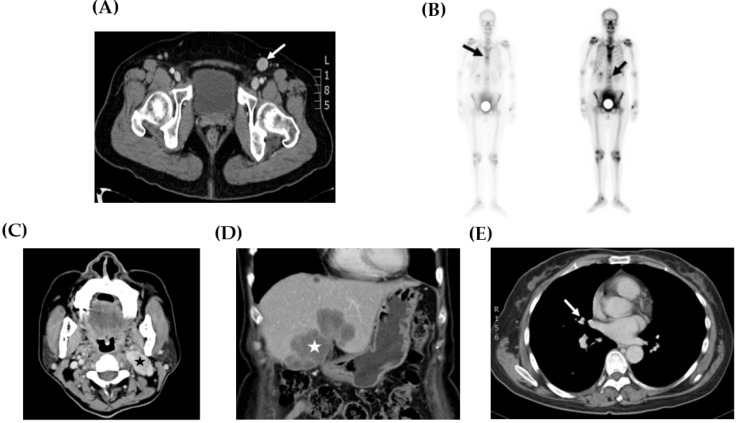
Representative image studies of recurrent tumors. (**A**) CT scan showed a 2.0 cm enlarged lymphadenopathy (arrow) in the left inguinal area (case 2). (**B**) Whole body bone scan demonstrated hot areas at the thoracic and lumbar spine (arrows) (case 1). (**C**) CT scan showed a 4.0 cm necrotic mass suggesting metastases at the left neck area (star) (case 5). (**D**) CT scan showed a 6.0 cm tumor at the lateral and medial segments of the liver (star) (case 5). (**E**) CT scan exhibited a 0.7 cm pulmonary nodule at the superior segment of the right lower lobe (arrow) (case 5).

**Figure 3 ijerph-16-00552-f003:**
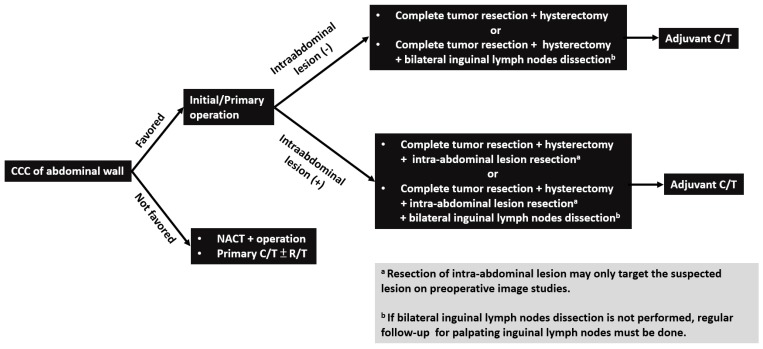
Flowchart of management for patients with clear cell carcinoma (CCC) of abdominal wall. *Note:* CCC, clear cell carcinoma; NACT: neoadjuvant chemotherapy; C/T: chemotherapy; R/T: radiotherapy.

**Table 1 ijerph-16-00552-t001:** Clinicopathologic characteristics of the six patients with clear cell carcinoma of the abdominal wall.

	Case 1	Case 2	Case 3	Case 4	Case 5	Case 6
Basic Characteristics						
Age (year)	52	56	52	56	55	45
BMI (kg/m^2^)	28.8	26.0	17.0	20.8	22.6	20.6
Parity	2	2	0	1	3	3
Previous Obs/Gyn surgeries	Cesarean section	Cesarean section	LSC left oophorectomy	Cesarean section	Cesarean section	Cesarean section
Time to onset (year) ^a^	19.0	33.0	4.0	21.0	24.0	20.0
Tumor size (cm)	17.5	6.5	7.0	12.0	12.5	4.8
Presenting symptoms	Lower abd. wall mass with ulceration	Lower abd. wall mass	LLQ mass	Lower abd. wall mass	Lower abd. wall mass	Painful lower abd. wall ulceration
Pretreatment serum CA-125 level (U/mL)	20.1	22.3	38.5	23.0	26.7	38.9
Preoperative CT scanning findings	1. A lower abd. wall tumor 2. A 6.5 cm left adnexal tumor	1. A lower abd. wall tumor 2. Small LN along right external iliac vessels 3. No definite focal lesion in the Gyn organs	1. A lower abd. wall tumor 2. No definite focal lesion in the Gyn organs	1. A lower abd. wall tumor with peritoneal involvement 2. No definite focal lesion in the Gyn organs	1. Multiple lower abd. wall tumors 2. No definite focal lesion in the Gyn organs	1. A lower abd. wall tumor 2. No definite focal lesion in the Gyn organs
Treatment Strategies						
Tumor excision	+	+	+	+	+	-
Gyn surgeries	TAH, BSO	TAH, BSO, omentectomy, BPLND	TAH, BSO	TAH, BSO, BPLND	TAH, BSO, omentectomy (after NACT)	**-**
Neoadjuvant (preoperative) C/T	-	-	-	-	+ ^b^	-
Primary C/T	-	-	-	-	-	+ ^c^
Adjuvant (postoperative) C/T	-	+ ^d^	+ ^e^	+ ^f^	+ ^g^	-
Radiotherapy	-	-	-	-	-	+ ^h^
Pathology						
Abd. wall tumor	CCC	CCC	CCC	CCC	CCC	CCC ^i^
Surgical margin	Free	Free	Free	Free	Involved	- ^i^
Co-existing endometriosis in abd. wall tumor^j^	-	-	+	-	+	-
Histological appearance showing endometrial stroma and glands in abd. wall tumor	-	-	-	-	-	-
Other primary tumor sites	-	-	-	-	-	-
CCC of gyn organs	Negative	Negative ^k^	Negative	Negative	Negative	-
Presence of suture granuloma	Yes	No	No	Yes	No	No
Intraabbdominal endometriosis	Adenomyosis Endometrioma	No	Adenomyosis	No	Adenomyosis	No

Note. BMI, body mass index; Obs/Gyn, obstetric and gynecologic; C-section, Cesarean section; LSC, laparoscopic; LLQ, left lower quadrant; TAH, total abdominal hysterectomy; BSO, bilateral salpingo-oophorectomy; BPLND, bilateral pelvic lymph node dissection; RSO, right salpingo-oophorectomy; NACT, neoadjuvant chemotherapy; CCC, clear cell carcinoma; C/T, chemotherapy; abd., abdominal; CA-125, cancer antigen 125; CT, computed tomography; LN, lymph node; + means “with” and – means “without”. ^a^ Time to onset was defined as the interval between the most recent Obs/Gyn surgeries and diagnosis of CCC of abdominal wall. ^b^ Five cycles of triweekly paclitaxel (175 mg/m^2)^ and carboplatin (AUC = 5) combined with bevacizumab (7.5 mg/kg) plus one cycle of gemcitabine (800 mg/m^2^) and carboplatin (AUC = 5). ^c^ Seven cycles of paclitaxel (175 mg/m^2^) and carboplatin (AUC = 5) plus one cycle of liposomal doxorubicin (40mg/m^2^) and carboplatin (AUC = 5). ^d^ Eight cycles of triweekly paclitaxel (175 mg/m^2^) and carboplatin (AUC = 5). ^e^ Six cycles of triweekly paclitaxel (175 mg/m^2^) and carboplatin (AUC=5). ^f^ The patient is under adjuvant chemotherapy now with triweekly paclitaxel (175 mg/m2) and carboplatin (AUC = 5). ^g^ Three cycles of gemcitabine (800 mg/m) and carboplatin (AUC = 5) combined with bevacizumab (7.5 mg/kg). ^h^ Three-dimensional conformal radiation therapy (3DRT) with 4400 cGy delivered in 22 fractions. ^i^ CCC was diagnosed via tumor biopsy without tumor resection, thus the involvement of surgical margin could not be assessed in this case. ^j^ Co-existing endometriosis refers to the endometriosis existing associated with CCC of the abdominal wall. ^k^ Metastasis to the right pelvic lymph node was noted, although there was no evidence of gynecologic organs involvement.

**Table 2 ijerph-16-00552-t002:** Clinical outcomes of the six patients with clear cell carcinoma of the abdominal wall.

	Case 1	Case 2	Case 3	Case 4	Case 5	Case 6
Recurrent/metastatic site	Inguinal LN, bone	Inguinal LN	No recurrence	NA ^a^	Abd. wall, inguinal LN, neck LN, liver, lung	Abd. wall, inguinal LN
Salvage treatment	-	Tumor excision	-	-	Chemotherapy ^b^ + Radiotherapy ^c^	-
DFS (m)	10	3	93	NA^a^	Progression	Progression
OS (m)	14	11	97	5	23	7
Current status	Recurrence	NED	NED	NED	DOD	Progression

*Note.* Abd., abdominal; LN, lymph node; NED, no evidence of disease; DOD, died of disease; NA, not available; DFS, disease-free survival; OS, overall survival. ^a^ The woman is under adjuvant chemotherapy. After removing the tumor with free margins, no evidence of disease is noted. ^b^ Cisplatin (35 mg/m^2^) on D1, D8, 5-FU (2600 mg/m^2^) on D1, D8, D15 and leucovorin (300 mg/m^2^) on D1, D8, D15 combined with bevacizumab (7.5 mg/kg) on D1 every three weeks for six cycles, and two cycles of doxorubicin liposomal (40 mg/m^2^) combined with bevacizumab (7.0 mg/kg). ^c^ Intensity-modulated radiotherapy (IMRT) with 5500 cGy was delivered in 25 fractions for metastatic lesion at the abdominal wall and inguinal.

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
