# Peer review of "Clear Cell Carcinoma of the Abdominal Wall as a Rare Complication of General Obstetric and Gynecologic Surgeries: 15 Years of Experience at a Large Academic Institution"

_ijerph, 2019, doi:10.3390/ijerph16040552_

Round 1
Reviewer 1 Report
I have read with interest the report of Drs Lai and colleagues. They are to be congratulated for a well written and concise manuscript with pertinent information. In general, I do believe this article merits publication, but I would encourage a handful of revisions to strengthen the manuscript. I have labeled these as minor (technical or semantic) and major (thematic) below
Minor
1 page 7 , line 2 Please explain why this patient did not receive additional treatment. (In other cases this is described)
2. There is mention that patients were followed up, and as a retrospective study I would not expect these to be necessarily standardized, but it would be good to know what the routine practice at you institution is.
3. Page 9 lines 7-8. I do not understand this sentence. It should be reworded.
4. Figure 3 is fascinating – but adds little to the report. It should be omitted. The manuscript reflects that mesh or flaps may be required.
Major
1. Avoid unsubstantiated claims (even when they are likely correct). Eg P9, line 31-32 How do you know that these tumors can “sometimes rapidly progress”? You cohort is small and most have a very long time interval between proposed initiation (gyn surgery) and event (tumor palpation) – therefore how can you tell if it grew quickly or slowly (the evidence would suggest slowly).
Eg 2 – There is no data presented to suggest that a 2 cm margin should be the goal and given the small number of patients (3 of which had recurrence) it would be hard to draw such a conclusion. Similarly – while your patients did not have identifiable tumor in the “normal appearing” gyn structures – 6 patients who have mixed results is a limited data set upon which to make the conclusion that hysterectomy can be omitted. Given the rarity of this condition it would seem more likely that a surgeon would encounter uterine disease with extra-uterine metastasis, than isolate extra-uterine disease – therefore it would seem more likely that if the hysterectomy was omitted that disease would be left behind. Remember, this series is selected for having abdominal wall disease in the known absence of other disease – which would not be the case prospectively.
2. Figure 4 suggests that observation is an option for these patients – but all patients who underwent observation ultimately recurred, while all patients who were NED had received some form of treatment. I do not think that even with the limited data you have that the option of observation is supported.
Author Response
Response to Reviewer 1 Comments
Minor revision
Point 1: Page 7, line 2 Please explain why this patient did not receive additional treatment. (In other cases this is described)
Response 1: Thank you for your comments. We have shown the reason that case 1 did not receive salvage treatment. (Please see page 8, lines 4-5)
Point 2: There is mention that patients were followed up, and as a retrospective study I would not expect these to be necessarily standardized, but it would be good to know what the routine practice at your institution is.
Response 2: Thank you for your comments. We have shown the routine postoperative surveillance at our institution. (Please see page 2, lines 85-89)
Point 3: Page 9 lines 7-8. I do not understand this sentence. It should be reworded.
Response 3: Thank you for your comments. We have rewritten the sentence. (Please see page 10, lines 7-8)
Point 4: Figure 3 is fascinating – but adds little to the report. It should be omitted. The manuscript reflects that mesh or flaps may be required.
Response 4: Thank you for your comments. We have removed the figure 3 and kept the description as your suggestion. (Please see page 10, line 41; page 12)
Major revision
Point 1: Avoid unsubstantiated claims (even when they are likely correct).
Eg 1– Page 9, line 31-32 How do you know that these tumors can “sometimes rapidly progress”? You cohort is small and most have a very long time interval between proposed initiation (gyn surgery) and event (tumor palpation) – therefore how can you tell if it grew quickly or slowly (the evidence would suggest slowly).
Eg 2 – There is no data presented to suggest that a 2 cm margin should be the goal and given the small number of patients (3 of which had recurrence) it would be hard to draw such a conclusion. Similarly – while your patients did not have identifiable tumor in the “normal appearing” gyn structures – 6 patients who have mixed results is a limited data set upon which to make the conclusion that hysterectomy can be omitted. Given the rarity of this condition it would seem more likely that a surgeon would encounter uterine disease with extra-uterine metastasis, than isolate extra-uterine disease – therefore it would seem more likely that if the hysterectomy was omitted that disease would be left behind. Remember, this series is selected for having abdominal wall disease in the known absence of other disease – which would not be the case prospectively.
Response 1: Thank you for your comments. We have corrected the manuscript. (Please see page 1, line 26, lines 34-35; page 4, table 1; page 7, lines 20-21; page 10, lines 31-33, lines 39-40, lines 44-50; page 11, lines 31-34; page 14, figure 3; page 15, lines 3-5)
Point 2: Figure 4 suggests that observation is an option for these patients – but all patients who underwent observation ultimately recurred, while all patients who were NED had received some form of treatment. I do not think that even with the limited data you have that the option of observation is supported.
Response 2: Thank you for your comments. We have corrected the manuscript. (Please see page 1, line 36; page 11, lines 9-16; page 14, figure 3; page 15, lines 6-7)
Reviewer 2 Report
A table containing the information on : (1) how many cases had malignant tissues and endometriotic lesions within the tumor, (2) Histological features showing endometrial stroma/gland and the tumor or (3) Additional primary site(s) of tumor---may be better for understanding cases.
Author Response
Response to Reviewer 2 Comments
Point 1: A table containing the information on:
(1) how many cases had malignant tissues and endometriotic lesions within the tumor, (2) histological features showing endometrial stroma/gland and the tumor or
(3) additional primary site(s) of tumor---may be better for understanding cases.
Response 1: Thank you for your comments. In table 1, there’s a column named “Co-existing endometriosis in abd. wall tumor”, that showed the presence of endometriotic lesion within the abdominal wall tumors in every case. Two more items: “Histological appearance showing endometrial stroma and glands in abd. wall tumor” and “Other primary tumor sites” have been added as your suggestions (Please see page 5, table 1).
Round 2
Reviewer 1 Report
The revisions have addressed my concerns and I think this manuscript merits publication.